# Drone-Based Community Assessment, Planning, and Disaster Risk Management for Sustainable Development

**Daniel Whitehurst** [1,*], **Brianna Friedman** [1], **Kevin Kochersberger** [2], **Venkat Sridhar** [3] and **James Weeks** [4]

1   Virginia Tech, Blacksburg, VA 24061, USA; fbrianna@vt.edu
2   Mechanical Engineering, Virginia Tech, Blacksburg, VA 24061, USA; kbk@vt.edu
3   Biological Systems Engineering, Virginia Tech, Blacksburg, VA 24061, USA; vsri@vt.edu
4   Development Monitors, LLC, Arlington, VA 22202, USA; jweeks@developmentmonitors.com
\*   Correspondence: dsw418@vt.edu

**Abstract:** Accessible, low-cost technologies and tools are needed in the developing world to support community planning, disaster risk assessment, and land tenure. Enterprise-scale geographic information system (GIS) software and high-resolution aerial or satellite imagery are tools which are typically not available to or affordable for resource-limited communities. In this paper, we present a concept of aerial data collection, 3D cadastre modeling, and disaster risk assessment using low-cost drones and adapted open-source software. Computer vision/machine learning methods are used to create a classified 3D cadastre that contextualizes and quantifies potential natural disaster risk to existing or planned infrastructure. Building type and integrity are determined from aerial imagery. Potential flood damage risk to a building is evaluated as a function of three mechanisms: undermining (erosion) of the foundation, hydraulic pressure damage, and building collapse due to water load. Use of Soil and Water Assessment Tool (SWAT) provides water runoff estimates that are improved using classified land features (urban ecology, erosion marks) to improve flow direction estimates. A convolutional neural network (CNN) is trained to find these flood-induced erosion marks from high-resolution drone imagery. A flood damage potential metric scaled by property value estimates results in individual and community property risk assessments.

**Keywords:** drone; aerial imagery; disaster risk management; classification; 3D modeling



## 1. Introduction and Motivation

Effective disaster risk management (DRM) requires accurate and up-to-date models [1,2]. In many cases, DRM and supporting information comes from multiple disaggregated and potentially out-of-date sources that may not reflect the currently built environment [2–4]. An increase in disaster risks is also being driven by population growth and rapid urbanization [5]. To encourage the use of an updated database that includes high-resolution features, we present a method based on small, low-cost drone imagery and adapted open-source tools to classify the built environment and use this information to model flood and other risks. Resource-limited communities will benefit from on-demand use of easily captured, low-cost, and high-resolution drone imagery to periodically update property and natural hazard risk information [6,7].

This article presents new analytical tools to conduct aerial imagery-based assessments for mitigating risk to existing and planned infrastructure. The goals of this work were: (1) to produce a DRM solution using low-cost drones for data collection, 3D cadastre modeling with the drone data, and disaster risk quantification for buildings; (2) to use drone imagery to expand upon existing SWAT analysis tools for producing flood models in areas with limited data; (3) to quantify building risk by combining flood models with building classification using drone imagery. While the methods presented in this paper apply to flooding due to this historically being a prime contributor to property damage [8], similar methods can be used to assess other risk sources. These risk sources could include

earth movement, such as erosion, or urban development that alters an existing natural hydrological response.

Accurate cadastre information, or datasets that define property boundaries and ownership, are frequently lacking in the developing world [9]. The lack of property titles can lead to conflict and insecurity, which, in turn, leads to instability for a country [9]. The Director of Land Affairs in the Democratic Republic of the Congo recently stated that "drones facilitate the real-time collection and rapid updating of land data, compared with traditional methods" [10]. A few accurate ground reference points visible in drone imagery are all that is required to complete a detailed cadastre for a community.

The ability to capture high-resolution imagery with a low-cost drone is matched to recent developments in Artificial Intelligence (AI) that can rapidly classify an area for features and objects of interest as well as create a reconstructed 3D environment for modeling purposes [11,12]. A fully convolutional neural network with Feature Pyramid Network architecture from the 2020 Open Cities AI Challenge hosted by the World Bank [13] can effectively classify buildings. The trained models from the challenge are available for download and we plan to further train these models for different locations. The resulting layer of identified features from the neural network can be used to build a flood model of an environment. OpenDroneMap is an efficient, open-source structure from motion (SfM) program used to produce 3D models from aerial imagery [14]. This software can additionally generate digital terrain maps (DTM) from the drone imagery. The outputs of these programs could be used to further classify buildings by condition with the use of observed roof type as a metric for construction quality.

Flood analysis using the Soil and Water Assessment Tool (SWAT) [15] is an efficient way to determine how flood events impact communities. SWAT accounts for terrain features, such as water flow to sub-basins caused by gradients, and land use including agricultural, developed, and undeveloped land. In the context of urban ecology, SWAT models can be used to show the beneficial effects of limited development to improve water uptake, which reduces flood height and volume.

We used the SWAT output as a baseline dataset for water damage potential along with the detection of erosion patterns in aerial imagery to provide a remotely sensed "ground truth" of water movement. Water marks, or erosion-caused ditches, can be correlated to slope to improve the accuracy of flow estimation and the potential for damage to buildings. The machine learning program Picterra [16] is used to find erosion patterns after an initial training dataset is used, which provides large-scale water movement evidence. Combined with a building classifier, risk is calculated using a weighted risk metric that accounts for building integrity/quality and the proximity to water-caused erosion.

## 2. Drone-Based Data Collection and Analysis

### 2.1. Motivational Forces in Disaster Risk Management

According to the World Bank, since 1980, more than two million people and over $3 trillion have been lost to disasters caused by natural hazards, with total damages increasing by more than 600% from $23 billion a year in the 1980s to $150 billion a year in the last decade [5]. In many developing countries, disaster risk is not contextualized at the community level during the project-planning process. An example is a school in Afghanistan funded by the World Bank, shown in Figures 1 and 2, that illustrates the disastrous impact natural hazards can have on infrastructure if not mitigated during the project planning process. The satellite images in Figure 1 show the location of a school before and after flooding of the Amu Darya river in Balkh Province, Afghanistan in April 2018. The photo in Figure 2 shows the extent of the damage caused to the school by flooding. This example underscores the need to provide donors, governments, and communities in developing nations access to low-cost data collection and analysis tools to assess and minimize disaster risk in order to protect lives and investments.

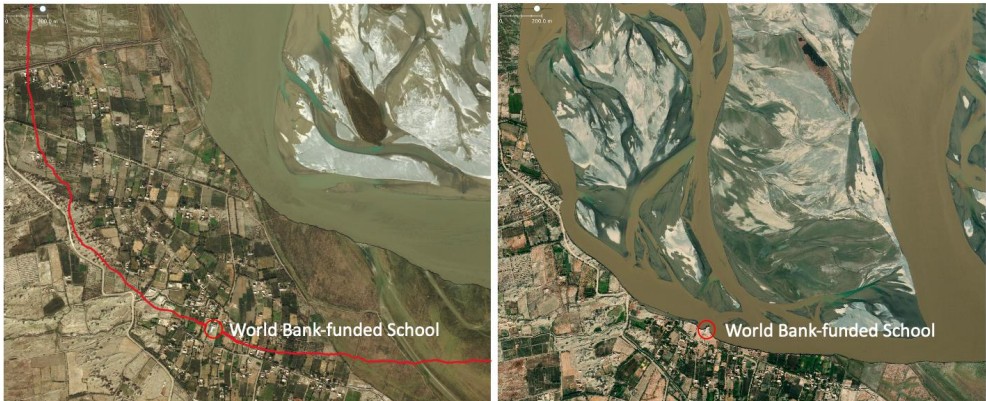

**Figure 1.** The location of a school in Afghanistan before and after a flood (Satellite imagery: ESRI). This flood of the Amu Darya river occurred in April 2018.

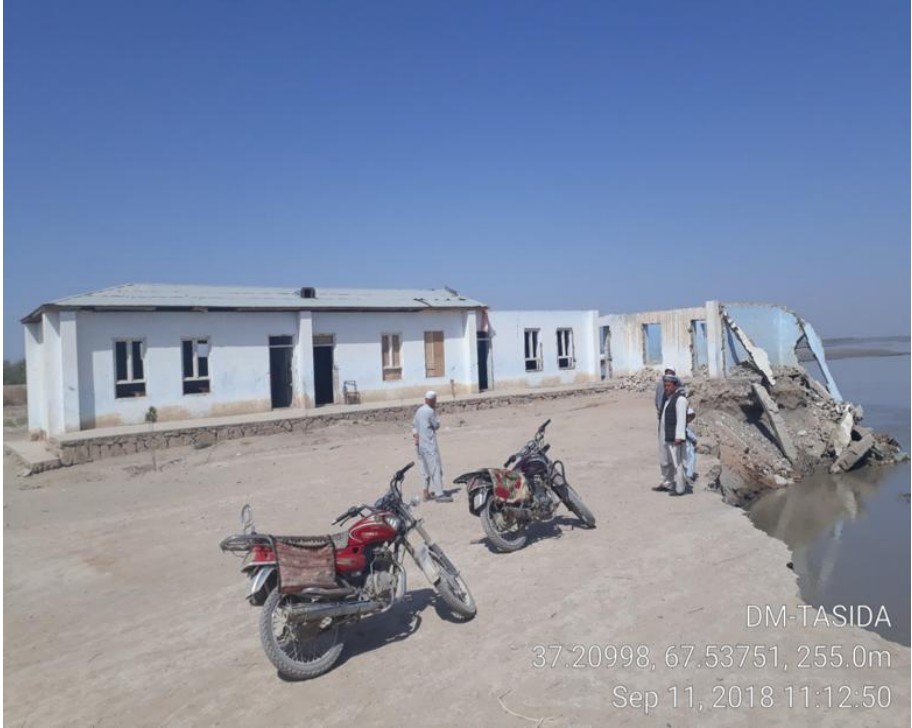

**Figure 2.** A school constructed in an Amu Darya river flood plain in Afghanistan and partially destroyed by flooding. This image was provided by Development Monitors LLC.

### 2.2. Down-Sized and Low-Cost Drone Technology

The drone market continues to evolve, with several small and low-cost drones now available on the consumer market. DJI is the world's largest consumer drone producer, with their products accounting for around 70 percent of the global consumer and enterprise drone market [17]. In addition to commercial drones for business, DJI also sells smaller and lower-cost drones, costing less than US $500. A comparison of the three smallest DJI drones is shown in Figure 3.

The main advantage of sub-250 gram drones is that a country's regulatory authority may recognize the reduced operational risk for the low-weight aircraft, making access to airspace easier [18]. Drones weighing less than 250 grams have inherently less kinetic energy, and therefore less injury and damage potential, making them appealing for flight operations in areas with higher population density.

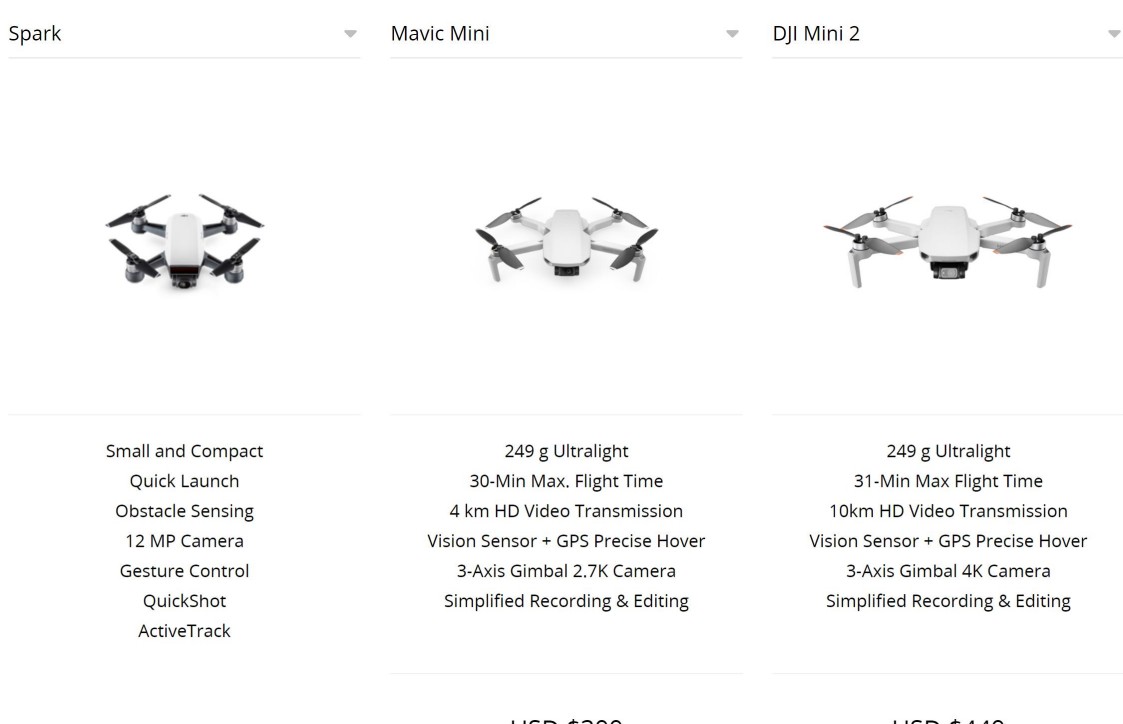

| Spark | Mavic Mini | DJI Mini 2 |
| --- | --- | --- |
| Small and Compact | 249 g Ultralight | 249 g Ultralight |
| Quick Launch | 30-Min Max. Flight Time | 31-Min Max Flight Time |
| Obstacle Sensing | 4 km HD Video Transmission | 10km HD Video Transmission |
| 12 MP Camera | Vision Sensor + GPS Precise Hover | Vision Sensor + GPS Precise Hover |
| Gesture Control | 3-Axis Gimbal 2.7K Camera | 3-Axis Gimbal 4K Camera |
| QuickShot | Simplified Recording & Editing | Simplified Recording & Editing |
| ActiveTrack | | |
| | USD $399 | USD $449 |

**Figure 3.** A comparison of the three smallest drones produced by DJI. This comparison tool is available on the DJI website [19].

The use of a small, low-cost, and expendable drone can enable the inexpensive collection of high-resolution aerial imagery. The Virginia Tech Unmanned Systems Lab designed and assembled a custom drone from inexpensive, off-the-shelf components. This drone weighs less than 250 grams and is shown on a scale in Figure 4. The drone is made from a custom carbon fiber frame and uses a flight controller running ArduPilot software. The imaging system consists of a Raspberry Pi Camera V2 connected to a Raspberry Pi Zero board. The camera is pointed downward in a fixed direction with a vibration isolated mount and is capable of capturing 8 megapixel imagery [20].

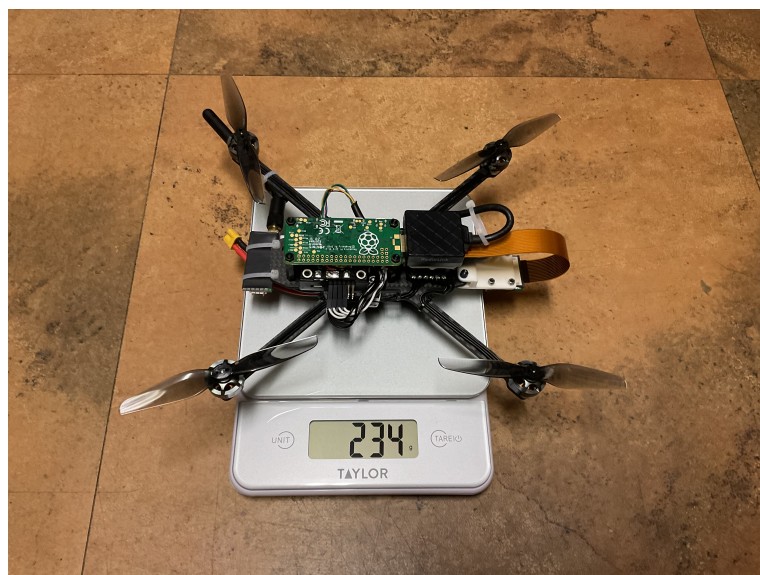

**Figure 4.** Our small, low-cost drone which weighs under 250 grams.

The drone is fully autonomous and flight plans are automatically generated using an application developed by the team. The planned mission covers a desired area while

acquiring imagery that meets the required overlap and ground sample distance criteria. The planned mission is uploaded to the drone using the open-source Mission Planner software [21]. The drone is equipped with a telemetry radio for in-flight communication with an Android application also developed by the team.

Once the mission has been created and sent to the drone, no manual user intervention is needed other than a command to start the mission. The drone autonomously flies its mission and collects high-resolution imagery before landing. Due to the lack of required user input, missions with this drone can be performed by minimally trained operators, thereby enabling this system to be used by locals in communities throughout the world. The flight controller includes fail-safe features where the drone will return to the home location and land in the event of a low battery or other issues during the mission. While the current drone does not include obstacle avoidance features, we hope to include these capabilities in a future version.

### 2.3. Machine Learning with Drone Imagery

Machine Learning techniques can be used on the high-resolution drone imagery in order to detect and classify objects and features. For our work, machine learning methods could be used to detect buildings, classify building roof type, and detect erosion patterns in the soil. Deep Neural Networks, or Artificial Neural Networks (ANN) with multiple layers, are considered one of the most powerful machine learning tools and have become very popular [22]. The Convolutional Neural Network (CNN) is one of the most popular deep neural networks [22]. "A Convolutional Neural Network is a Deep Learning algorithm which can take in an input image, assign importance (learnable weights and biases) to various aspects/objects in the image, and be able to differentiate one from the other" [23]. CNNs have achieved excellent performance in machine learning problems, especially applications that deal with image data, computer vision, and in natural language processing (NLP) [22]. CNNs are composed of multiple layers of artificial neurons, which are mathematical functions to output an activation value by calculating the weighted sum of multiple inputs [24]. The behavior of these neurons will be defined by their weights and they are able to pick out various visual features when the pixel values of an image are used as an input to the neural network [24]. CNNs can be "trained" using labeled input data to update the weights of its neurons in order to improve performance [24]. A test dataset is then used to verify the accuracy of the CNN on data it has not seen before [24].

Many open source machine learning libraries exist for Deep Learning with CNNs. Two of the most popular open source machine learning libraries are TensorFlow [25] and PyTorch [26]. In addition to these open source libraries, proprietary software systems are also available. For part of our work, the proprietary software Picterra was used for creating machine learning models. Picterra enables users to build and train detectors quickly through their intuitive online interface [16]. Although Picterra does not provide information on the model structure, the results of these detectors can be visualized and shared online as well as integrated with other platforms [16]. Picterra includes many pre-trained models as a starting point for training [27] and has been used for many different applications [28], including building detection [29].

## 3. The Soil and Water Assessment Tool (SWAT) in Urban-Scale Flood Modeling

SWAT is a river basin or watershed scale model that was created to predict the effect that land management practices have on water, sediment, and agricultural chemical yields over a long period of time [30]. The ability to model recharge and runoff makes SWAT an ideal tool to understand the dynamic behavior of water at the confluence of urban, forested, and agricultural areas frequently present in lower income communities. A SWAT model is based on the water balance equation (Equation (1)) [30].

$$SW_t = SW_0 + \sum_{i=1}^{t} \left( R_{day} - Q_{surf} - E_a - w_{seep} - Q_{gw} \right) \tag{1}$$

"where $SW_t$ is the final soil water content (mm $H_2O$), $SW_0$ is the initial soil water content on day $i$ (mm $H_2O$), t is the time (days), $R_{day}$ is the amount of precipitation on day $i$ (mm $H_2O$), $Q_{surf}$ is the amount of surface runoff on day $i$ (mm $H_2O$), $E_a$ is the amount of evapotranspiration on day $i$ (mm $H_2O$), $w_{seep}$ is the amount of water entering the vadose zone from the soil profile on day $i$ (mm $H_2O$), and $Q_{gw}$ is the amount of return flow on day $i$ (mm $H_2O$)." [30].

Created in the early 1990s, SWAT is a very popular hydrology model used in hundreds of published journal articles, many of which explore urban ecology—[31–33] just to name a few. In [31], the authors explore how changes in land use affect hydrology in the River Njoro watershed in Kenya. Baker et al. [31] created three land-use maps over a 17-year period, ran a SWAT simulation for each one, and found that land-use changes occurring in their study area (i.e., deforestation, increase in cropland) increased surface runoff and decreased groundwater recharge.

There are many hydrology tools available and it can be difficult to choose the right one for any particular study area. In [34–37], the authors compared SWAT to other commonly used hydrology models and found SWAT to be the best continuous model for a data-scarce area. In data-scarce areas, it can be impossible to calibrate SWAT. In [38–40], the authors compared calibrated and uncalibrated SWAT models to measured stream flow data and found that, in many situations, SWAT can be run without calibration. For example, [38] explores the importance of calibrating SWAT in the Santa Cruz River Watershed for forecasting absolute and relative changes in stream-flow by comparing the outputs of three SWAT models with varying degrees of calibration. The three degrees of calibration are: uncalibrated, outlet-calibrated (most commonly used), and spatially calibrated. Using the Percent Bias and Nash-Sutcliffe Efficiency, the observed flow was compared to the simulated flow on a monthly time scale. Although the uncalibrated model performed the worst, calibration did not affect the relative change in stream-flow due to precipitation or temperature. Niraula et al. [38] found that if relative change is the factor of importance, results from an uncalibrated model are sufficient. We are interested in comparing surface runoff at different points in our study area, relative to the average surface runoff throughout the study area, so relative change is the factor of importance.

The main variable of interest for our study is surface runoff. "Surface runoff, or overland flow, is flow that occurs along a sloping surface" [30]. SWAT has two methods for estimating surface runoff: the SCS Curve Number and the Green & Ampt Infiltration. The Green & Ampt Infiltration method was not an option for this study because it requires sub-daily precipitation data, which was unavailable for the study area. The SCS Curve Number method is an empirical model created after over 20 years of studies exploring rainfall–runoff relationships from small rural watersheds across the US and is computed using Equation (2) [30].

$$Q_{surf} = \frac{\left(R_{day} - 0.2S\right)^2}{(R_{day} + 0.8S)}$$
$$S = 25.4\left(\frac{1000}{CN} - 10\right)$$

(2)

where $Q_{surf}$ is the accumulated surface runoff/excess rainfall (mm $H_2O$), $R_{day}$ is the daily rainfall depth (mm $H_2O$), and CN is the daily curve number which is a function of soil properties and land use. To determine CN, each soil type is placed into one of four categories based on the soil's infiltration characteristics. The U.S. Department of Agriculture provides tables to determine CN [41].

## 4. Data Collection and Registration

### 4.1. Cadastre Systems and Data

Municipalities in developed countries use a wide variety of cadastre systems to document property ownership. A good example is from Montgomery County, Virginia

(Figure 5). In many areas, particularly in the United States, cadastral information can be found and downloaded freely online.

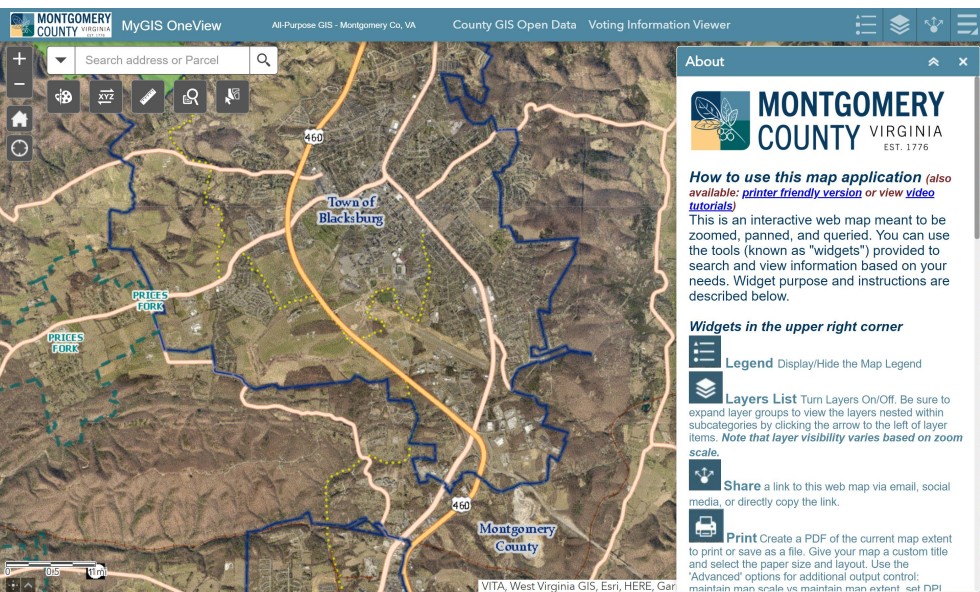

**Figure 5.** Sample cadastre system used in Montgomery County, VA, USA [42].

For the Commonwealth of Virginia, cadastral information is openly available on the Open Data Portal created by the Virginia Department of Transportation [43]. For our purposes, shapefile, or GeoJSON formats of cadastral data can be displayed in GIS software such as QGIS [44] as shown in Figure 6. The figure shows data and satellite imagery for Kentland Farm, which is a farm that was acquired by Virginia Tech for research purposes. While cadastre data is openly available for much of the United States, this is not the case in many developing countries [9].

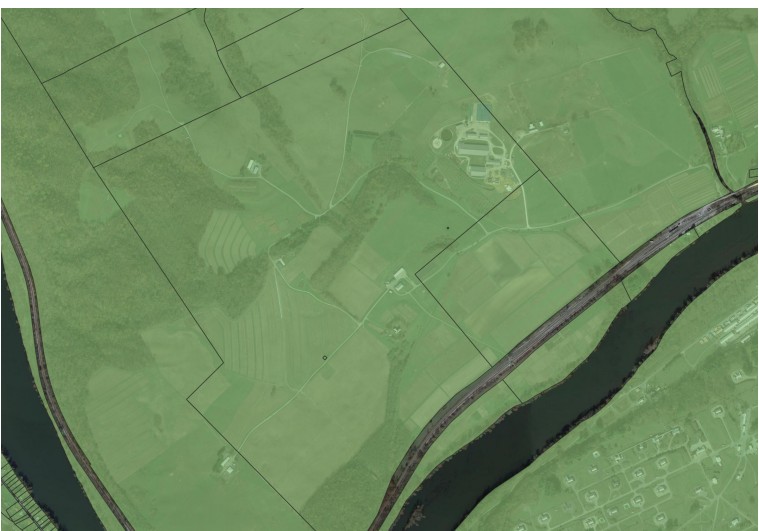

**Figure 6.** Property boundary data displayed in QGIS for the Virginia Tech Kentland research farm.

### 4.2. Natural Disaster Risk Applications and Data

Considerable effort in creating targeted and easy-to-implement disaster risk management (DRM) tools has resulted in a range of options for community planners. A good example is the Open HAZUS-MH tool [45] developed by the US Federal Emergency Management Agency (FEMA). This open-source risk modeling tool was developed for multi-hazard risk assessment in the US. As part of a World Bank project in Afghanistan,

Development Monitors created a natural disaster risk mapping plug-in for the open-source GIS application QGIS. The plug-in leverages available high-resolution imagery, multi-hazard natural disaster risk data, and a machine learning model to detect infrastructure features in order to create community-level disaster risk maps like the one shown in Figure 7. Multi-hazard risk data at the national and state levels are available to the general public from many sources, including FEMA, VFRIS, and others.

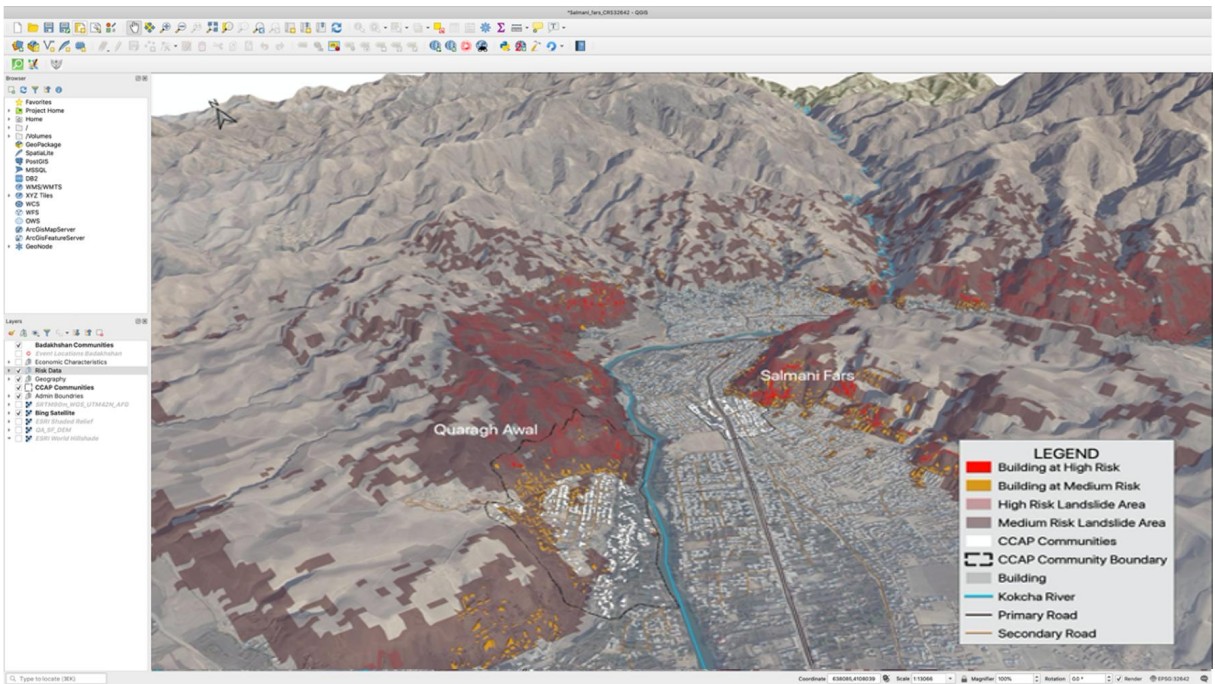

**Figure 7.** 3D landslide risk map of two communities in Badakhshan province, Afghanistan. This image was provided by Development Monitors LLC.

### 4.3. 3D Modeling

Many different approaches are being explored for 3D visualization and 3D cadastral mapping and modeling. This includes the 3D cartographic visualization of a historical topographic object [46]. The use of surveyed ground control points, real-time kinematic positioning data, and Structure from Motion algorithms enable the generation of detailed 3D models of objects and their surroundings with accuracy of up to several millimeters [46]. Models like this could be transferred and shared on public geospatial databases. Research is also being done on the use of modern game engines with geospatial data to create immersive virtual 3D environments for geographic visualization [47]. Additionally, significant research has been performed on the intersection of 3D cadastre data and building information modeling (BIM), which can provide a rich repository of legal and physical datasets in a common environment [48].

Using our drone system, we are able to collect high-resolution aerial imagery of properties. The collected imagery can then be post-processed to produce 3D reconstructions of an area. The open source OpenDroneMap software can be used to produce 3D textured models, point clouds, georeferenced orthorectified images, and georeferenced Digital Elevation Models from aerial imagery [14]. An example of a 3D reconstruction produced from aerial imagery collected by a small, lost-cost drone is shown in Figure 8. While the georeferenced models can be produced using geotagged images, surveyed ground control points are needed to produce more accurate location data. Once the 3D models have been created, these can be combined with cadastre and risk data in order to further contextualize this information.

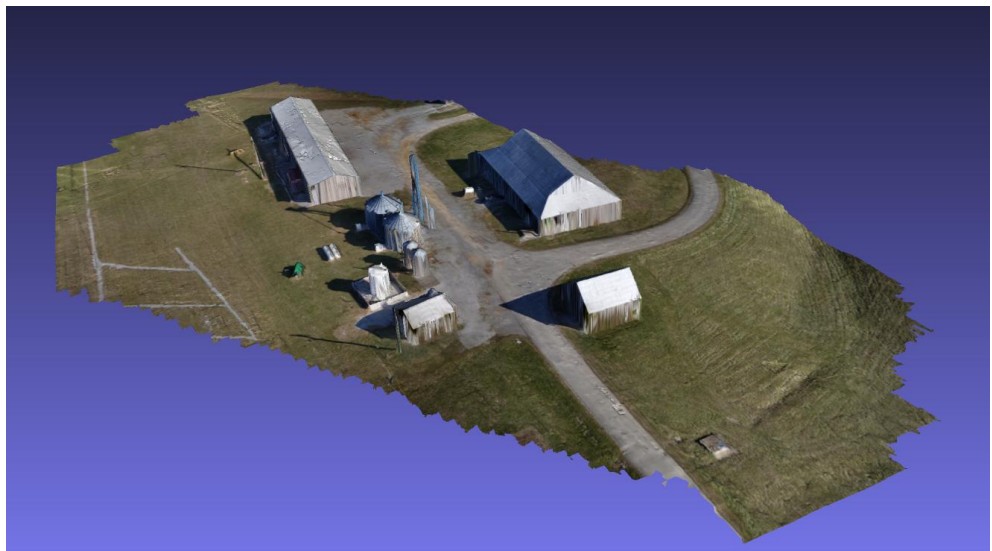

**Figure 8.** A 3D reconstruction produced from drone imagery captured at the Virginia Tech Kentland research farm.

### 4.4. Contextualizing 3D Models with Cadastre and Risk Data

In developing countries, drones coupled with a real-time kinematic (RTK) base station can enable communities to accurately create land registries [10]. In more developed countries, available cadastre and risk data can contextualize 3D models for mitigating risk at the property and building level.

After drone imagery has been processed by OpenDroneMap, georeferenced outputs can be aligned with and overlaid on top of property boundary data. The georeferenced orthorectified image, or orthophoto, of a building at Kentland Farm is shown in Figure 9. This orthophoto was overlaid on top of base satellite imagery and the property boundary data using QGIS. The georeferencing was performed using the GPS data from the drone for each of the individual images. While the image aligns reasonably well with the satellite imagery, a GPS base station with ground control points would improve the georeferencing accuracy. Once the orthophoto and 3D model data have been aligned with the cadastre data, available risk data can contextualize this at the property and building levels to show the specific properties at risk. Disaster risk information can be highlighted within the property boundary and matched to specific portions of the 3D building data to clearly show how the buildings may be impacted.

Additional outputs of the drone image processing include DTMs. These can be produced by the open source OpenDroneMap software as well as many other proprietary software options. The use of a DTM as one of the inputs for risk assessment is discussed further in the following section.

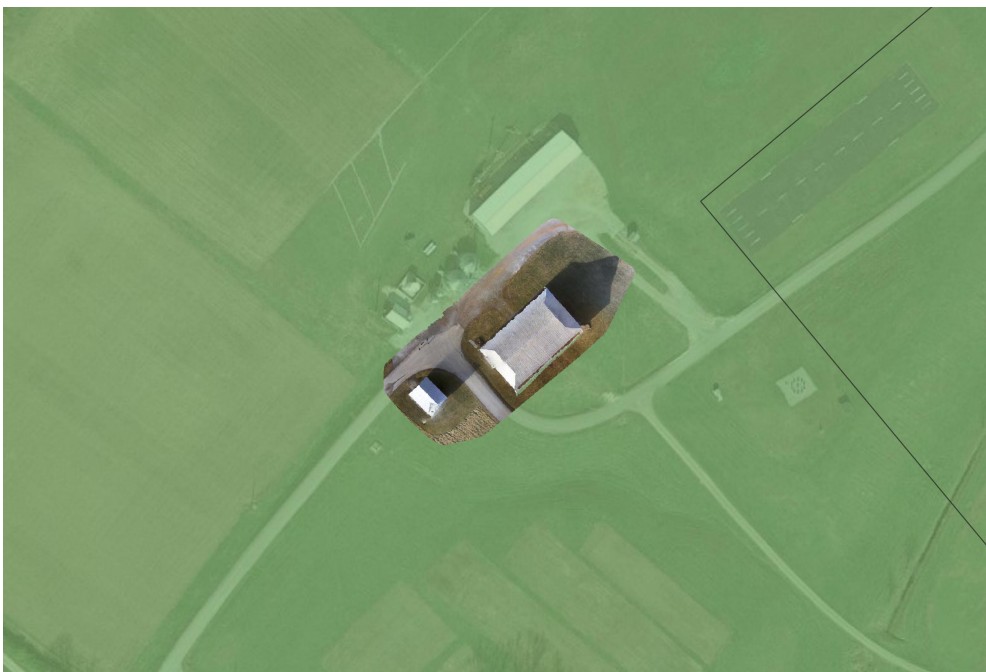

**Figure 9.** A 3D reconstruction orthophoto produced from drone imagery captured at Kentland Farm overlaid on satellite imagery and cadastre data in QGIS.

## 5. SWAT Modeling and Hydrology Modeling

*5.1. Dzaleka Refugee Camp Data Collection and Initial Flood Model*

Located in Dowa, Malawi, Dzaleka Refugee Camp is home to 41,000 refugees from the Democratic Republic of Congo, Rwanda, Burundi, Ethiopia, and Somalia [49]. When they arrive, many refugees build a house out of concrete and home-made bricks created from dug-up clay. With little urban planning and poor building materials, refugees' homes encounter flood damage, and many homes collapse, forcing refugees to build houses over and over again. In some cases, a collapsed house will also be abandoned by the owners, taking up precious space in an over-populated camp that was originally built for 10,000 people [49].

Water events can cause buildings at Dzaleka to collapse in three different ways: foundation undermining (Figure10a), hydraulic pressure damage (Figure10b), and roof collapse due to water load (Figure10c). Foundation undermining occurs from water running along the dirt around the base of a structure. Over time, the dirt will wear away, reducing the integrity of the structure's foundation. Hydraulic pressure damage occurs when water running along the surface applies pressure to the walls of a building causing the walls to collapse. Roof collapse due to water load occurs when the weight of rainfall is larger than the weight the roof can support.

This study explores the risk of foundation undermining (Figure10a) throughout Dzaleka Refugee Camp by using drone imagery as an input for SWAT and a machine learning program. SWAT is a small watershed model used to simulate the quantity of water over time [15]. SWAT requires four datasets: digital terrain model (DTM), land use map, soil type map, and weather data. Both the DTM and land use map were created using drone imagery.

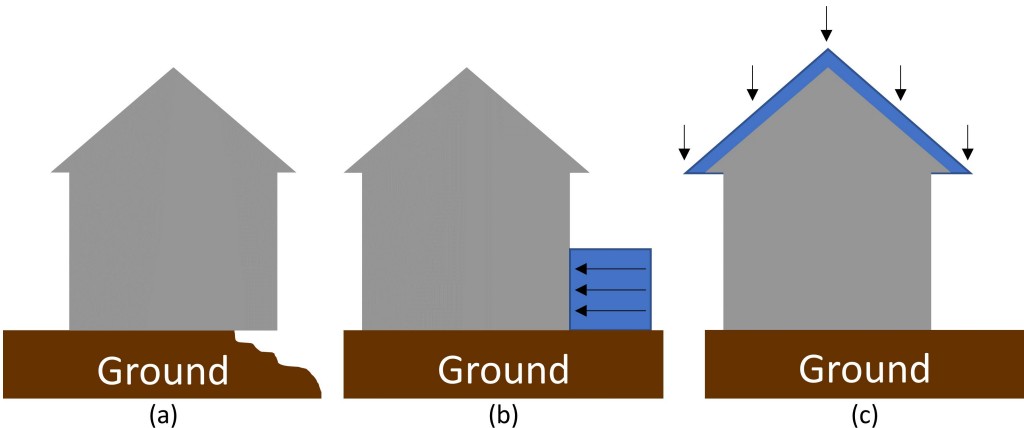

**Figure 10.** Buildings at Dzaleka Refugee Camp can collapse from water-related events in three different ways: (**a**) foundation undermining, (**b**) hydraulic pressure damage, and (**c**) roof collapse from water load.

In March 2020, the African Drone & Data Academy completed a mapping mission over Dzaleka Refugee Camp (4.6 km$^2$) using a Parrot Disco drone with a Parrot Sequoia camera on-board. The mission was flown 120 m above ground level with 70% image overlap. The 2603 RGB images were stitched together in Pix4D Mapper to create a digital terrain model (DTM) (Figure 11a) and RGB orthomosaic (Figure 11b), both of which have a resolution of 3.5 cm. Both the DTM and RGB orthomosaic were resampled to 20 cm resolution to reduce the processing power required. ArcGIS object classification was completed on the RGB orthomosaic to create a land-use map (Figure 11c) based on four classifications: urban, bare, shrubland, and row crops. The 20 cm resolution DTM and 20 cm resolution land-use map were then used as inputs for SWAT.

The soil-type map was created using the Harmonized World Soil Database (HWSD), which combines regional and national updates of worldwide soil information [50]. In HWSD, Malawi's soil data are from the Soil and Terrain database, which was initiated by the Food and Agricultural Organization of the UN in 1986 [51]. The soil data have a resolution of 1 km, with many of the data points being "filled in" through interpolation. All of Dzaleka Refugee Camp was classified as the same soil type with a dominant soil group of Lixisol.

The weather data were provided by the National Centers for Environment Prediction Climate Forecast System Reanalysis (CFSR) which has a horizontal resolution of 38 km and provides daily values for precipitation, temperature, wind, relative humidity, and solar radiation from 1979–2013 [52]. Fuka et al., 2014 compared the CFSR flood model output to the flood model output using the closest weather station for five different watersheds, [52] found that CFSR generally has equal to or more accurate streamflow predictions. The CFSR weather data were for a point 15.9 km northwest of the camp located at −13.582 S, 33.750 E. SWAT used the first four years as a "warm-up".

SWAT created 34 subbasins and 135 hydrological response units (HRU) for the study area. Figure 12 shows the stream network (white lines), subbasins (black outlines), and HRUs (colors). An HRU is a unique combination of subbasin, slope, land use, and soil type. Because the entire study area only has one soil type, for our case, an HRU is a unique combination of subbasin, slope, and land use.

Monthly accumulated surface runoff was used to correlate flood events to undermining. Surface runoff is the height of water running along the surface. This water neither infiltrates the soil nor evaporates. From 1983–2013, the month with the greatest rain accumulation was January 2013, totaling 714 mm of rain. The total generated surface runoff by HRU in January 2013 is shown in Figure 13a. The surface runoff varies drastically throughout Dzaleka, ranging from 265.3 mm to 479.5 mm. Because of this large range, it is clear that some areas of the camp are less impacted by rainfall than others. Instead of studying the worst event in recent history, the 10% surface runoff exceedance probability was analyzed, and again the study area experiences a large range of surface runoff. Figure 13b is the

10% surface runoff exceedance probability for Dzaleka Refugee Camp. The surface runoff ranges from 54.6 mm to 181.3 mm. Comparing surface runoff during the worst recorded flood event (Figure 13a) and the 10% exceedance probability event (Figure 13b), it is clear that the amount of surface runoff in a particular HRU relative to the entire study area does not change.

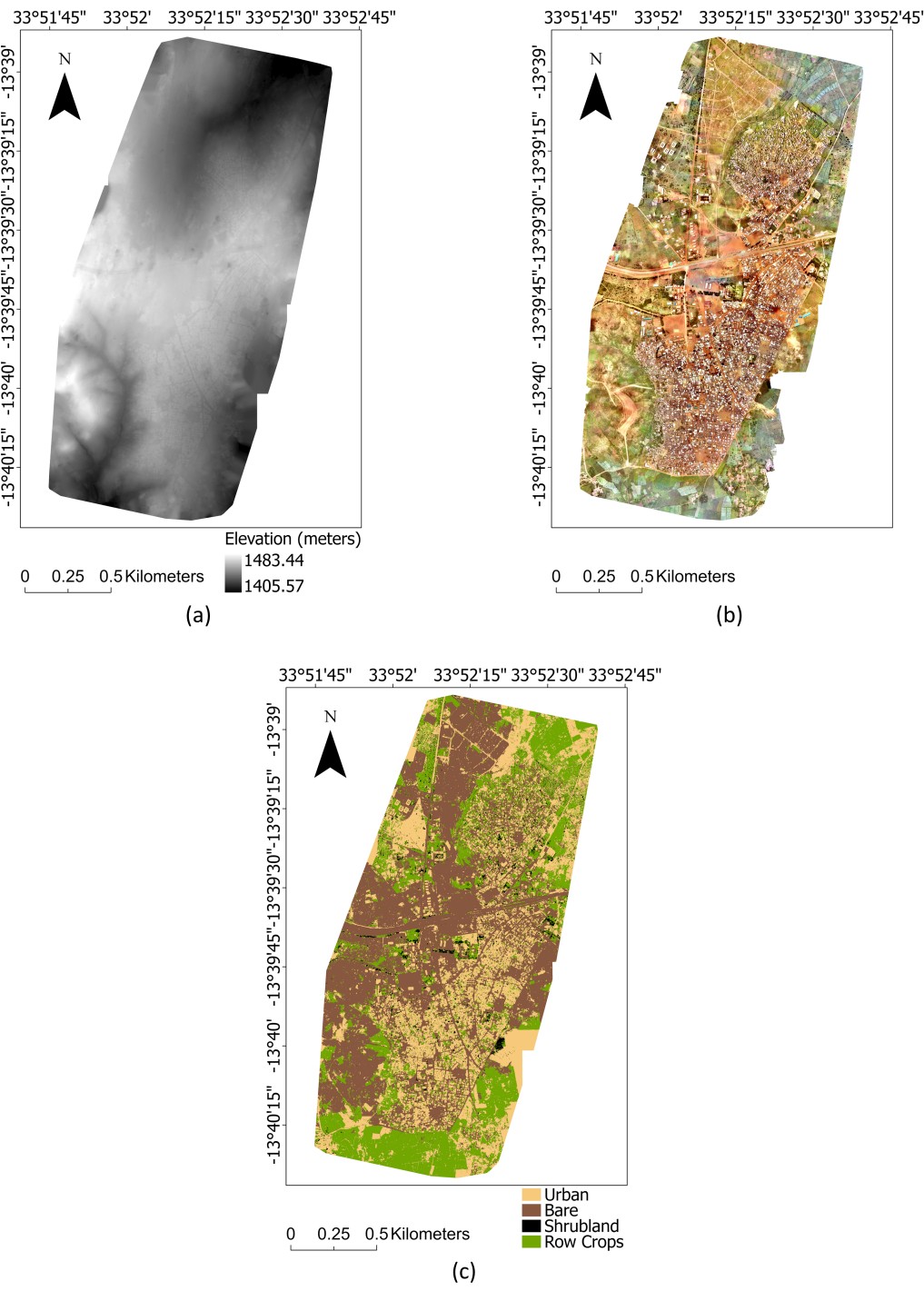

**Figure 11.** (**a**) Digital terrain model (DTM), (**b**) orthomosaic, and (**c**) land use map. The DTM and land-use maps were created by stitching 2603 images of Dzaleka Refugee Camp together in Pix4D Mapper. The land-use map was created by using ArcGIS Object Classification on the orthomosaic (Figure 11b). The study area contains four different land classifications: urban (tan), bare (brown), shrubland (black), and row crops (green).

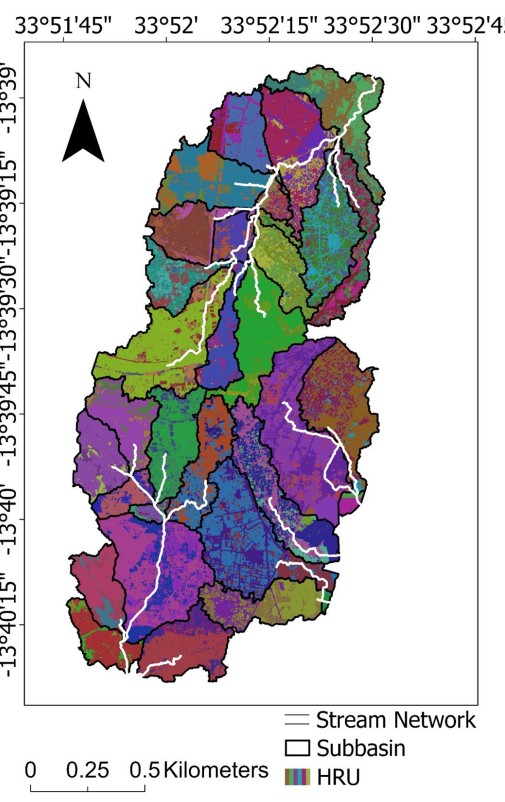

**Figure 12.** The initial SWAT output contains 34 subbasins (black outlines) based on the stream network (white lines) and 135 HRUs (colors).

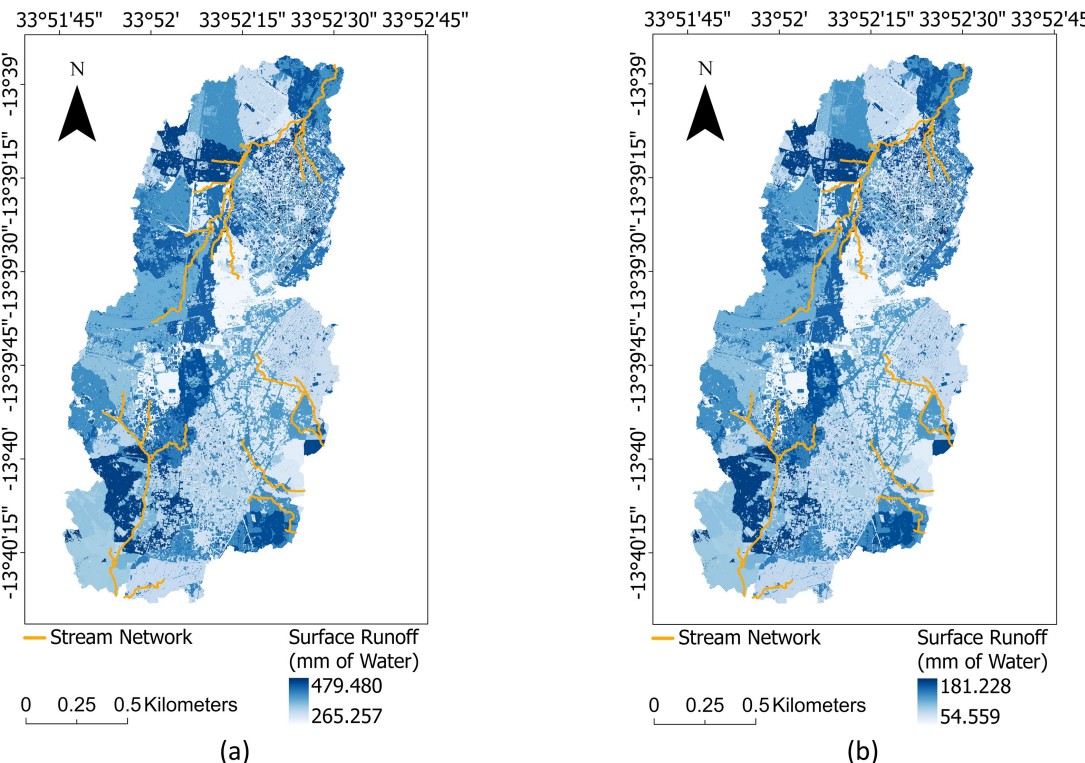

**Figure 13.** (**a**) Millimeters of water surface runoff accumulated in January 2013; (**b**) Millimeters of water surface runoff for a 10% exceedance event.

This is very useful information, but SWAT is only as powerful as the data provided to it. Although the DTM and land use map used for our model are incredibly high in resolution, the weather and soil data are not. Dzaleka also has no data that can be used for validation (i.e., measured streamflow). To improve the model, extra information from the drone imagery was extracted and then combined with the SWAT analysis.

### 5.2. Drone Imagery Used to Enhance Initial Flood Model

As previously mentioned, drone imagery resolution is very high (in our case 3.5 cm), which can provide valuable information about flood patterns. The left image in Figure 14a was captured during the mapping mission. This image is enlarged on the right of Figure 14a. In the enlarged image, there are a lot of erosion patterns (dark lines in the dirt). Some of these erosion patterns can provide even more information about flooding habits at the Dzaleka Refugee Camp. Instead of manually finding erosion marks, the proprietary machine learning program Picterra [16] was trained and used to detect erosion marks. Out of the available pre-trained models in the Picterra library, cracks are the most similar to erosion patterns. As a result, a pre-trained base model for cracks was used as the starting point for model training. Portions of the drone imagery were annotated within Picterra for training. These training labels enabled Picterra to produce a model which could be tested on different portions of the imagery. Figure 14b is an example of an erosion pattern found using the Picterra model.

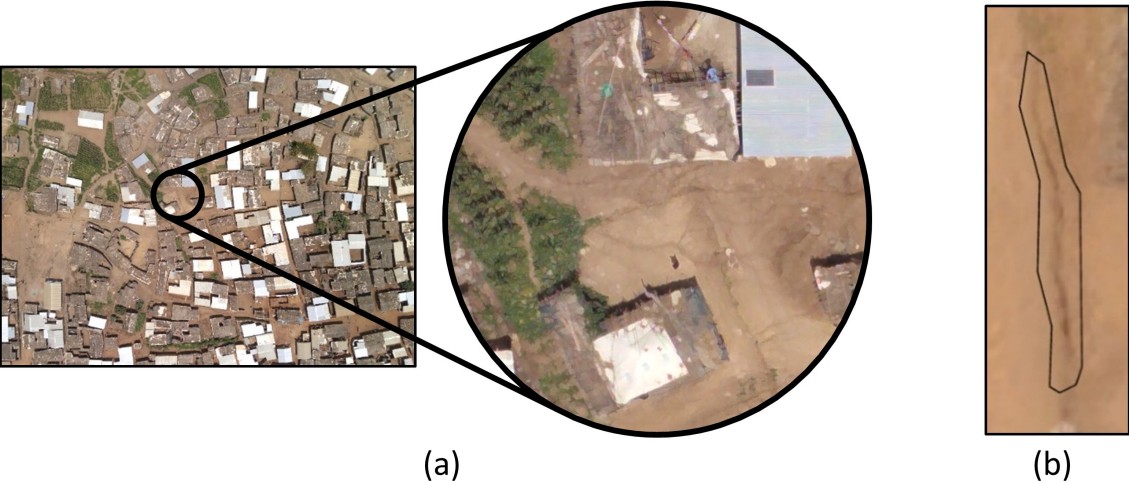

(a)  (b)

**Figure 14.** Picterra was used to find erosion patterns in the drone imagery. (**a**) An example of a drone captured image at Dzaleka Refugee Camp along with an emphasis on erosion patterns (dark lines) that Picterra was trained to find. (**b**) An example of a Picterra found erosion pattern after training.

Each found erosion pattern was converted into a line (purple in Figure 15a) and theta ($\theta$ in Figure 15a) was calculated. Theta was compared to a flow direction raster created using the D8 method on the DTM. This was done to determine which found erosion patterns were caused by water flow. The D8 method compares the elevation of a pixel to the elevation of all eight of its neighbors and then determines the direction in which water will flow. Each of the eight possible directions are given a number, 1, 2, 4, 8, 16, 32, 64, and 128 (see Figure 15b). If the erosion pattern line was in the same direction as the flow, then the mark was kept and considered to be a water-caused erosion pattern (WCEP). Table 1 provides the range of angles that correspond to each flow direction.

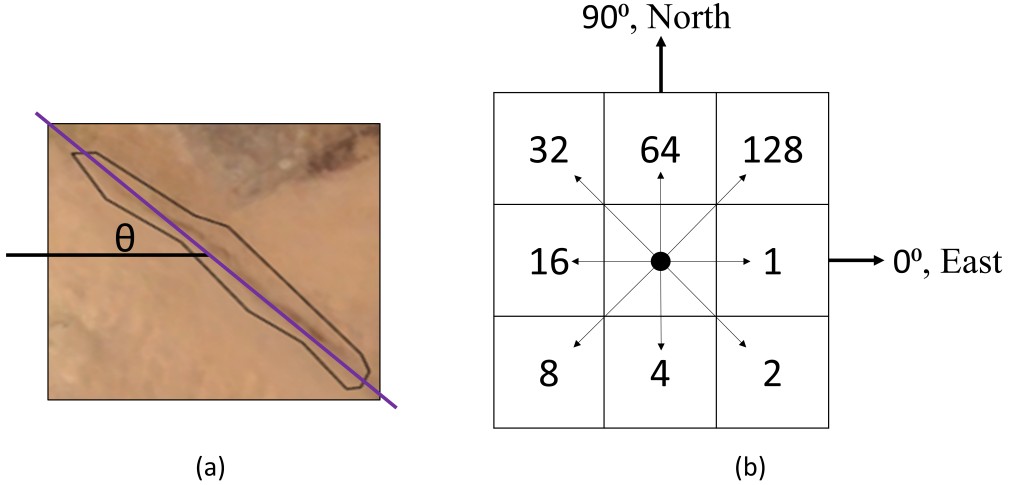

**Figure 15.** (**a**) To compute the angle of each found erosion pattern, each polygon was converted into a line (purple) and then theta was calculated; (**b**) Flow direction definition for the D8 method.

**Table 1.** Range of angles that correspond to each flow direction.

| Flow Direction | Range of Degrees |
|:---:|:---:|
| $1, 16$ | $-45 \rightarrow 45$ |
| $2, 32$ | $-90 \rightarrow 0$ |
| $4, 64$ | $-90 \rightarrow -45, 45 \rightarrow 90$ |
| $8, 128$ | $0 \rightarrow 90$ |

Each WCEP is not only providing information about that specific location but also about the immediate surrounding areas. A 10m buffer was applied to each WCEP to include the surrounding areas. There cannot be a WCEP in a location where there is a house because erosion patterns are only found in bare earth, so the 10m buffer enables structures close to WCEP to be included in the affected area. Although this was done for the entire study area, Figure 16e shows the found erosion patterns with a 10 m buffer (black and purple) and which portions of the erosion patterns are water caused (purple) for a small sample of the study area.

Locations of WCEPs are high undermining risk areas. A more comprehensive flood model was created by combining the WCEPs map (purple in Figure 16e) and the 10% exceedance probability for surface runoff (Figure 13b). The 10% exceedance probability for surface runoff (Figure 13b) was normalized from 0 to 1, and then each dataset was given a weight of 0.5 and added together. The created comprehensive flood model is shown in Figure 17; dark blue represents the highest flood-risk areas. By creating a comprehensive flood model, we are able to include ground truthed data (i.e., found WCEPs).

*5.3. Analysis of a Small Study Area within Dzaleka Refugee Camp*

To further explore the output data, a small study area of Dzaleka Refugee Camp was analyzed. Figures 16a–f are clips of the same small study area through different steps of the analysis. Figures 16a,b are the DTM and land use map, respectively. The elevation of this study area varied by 20 m and the study area contains all four different land-use classifications that are found at Dzaleka Refugee Camp. The normalized surface runoff for 10% exceedance probability produced by SWAT (Figure 16c) portrays how land use affects surface runoff. The patch of row crops (in the north and 0.15 km from the west border) has significantly lower surface runoff than the surrounding areas, which are urban or barren.

Figure 16d shows the orthomosaic of the study area along with black outlines of all the found erosion patterns. The erosion patterns were then given a 10 m buffer and compared to flow direction (Figure 16e). The comprehensive flood model (Figure 16f) is the

combination of the initial flood model (Figure 16c) and the WCEP (purple in Figure 16e). The comprehensive flood model allows for areas in one HRU to be distinguished from each other. Instead of the entire mid-west of the study area having a high flood risk (as in Figure 16c), the southern section of the mid-west is considered to have a higher flood risk because of the WCEPs there.

The comprehensive flood model is incredibly useful in future urban planning. Areas where surface runoff is high because of elevation change should be used for agriculture and not buildings, drainage systems should be implemented in high-risk areas, and houses built in high-risk areas should have a concrete foundation. Although the method of combing SWAT and machine learning still needs more verification, it has the potential to be applied to other data-scarce communities.

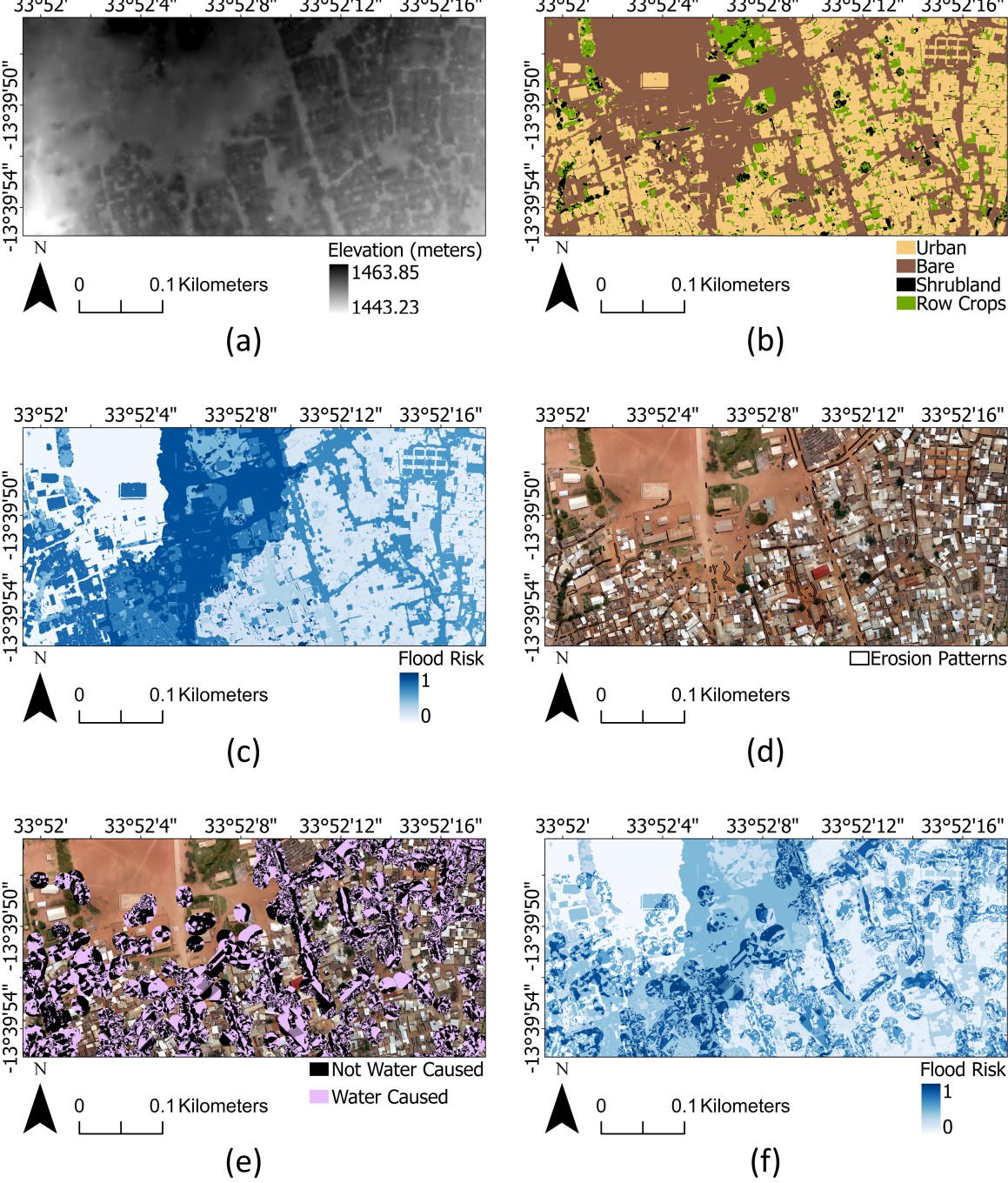

**Figure 16.** A small study area of Dzaleka Refugee Camp through flood analysis processing: (**a**) DTM; (**b**) land use map; (**c**) initial flood model; (**d**) erosion patterns; (**e**) erosion patterns compared to flow direction; (**f**) comprehensive flood model.

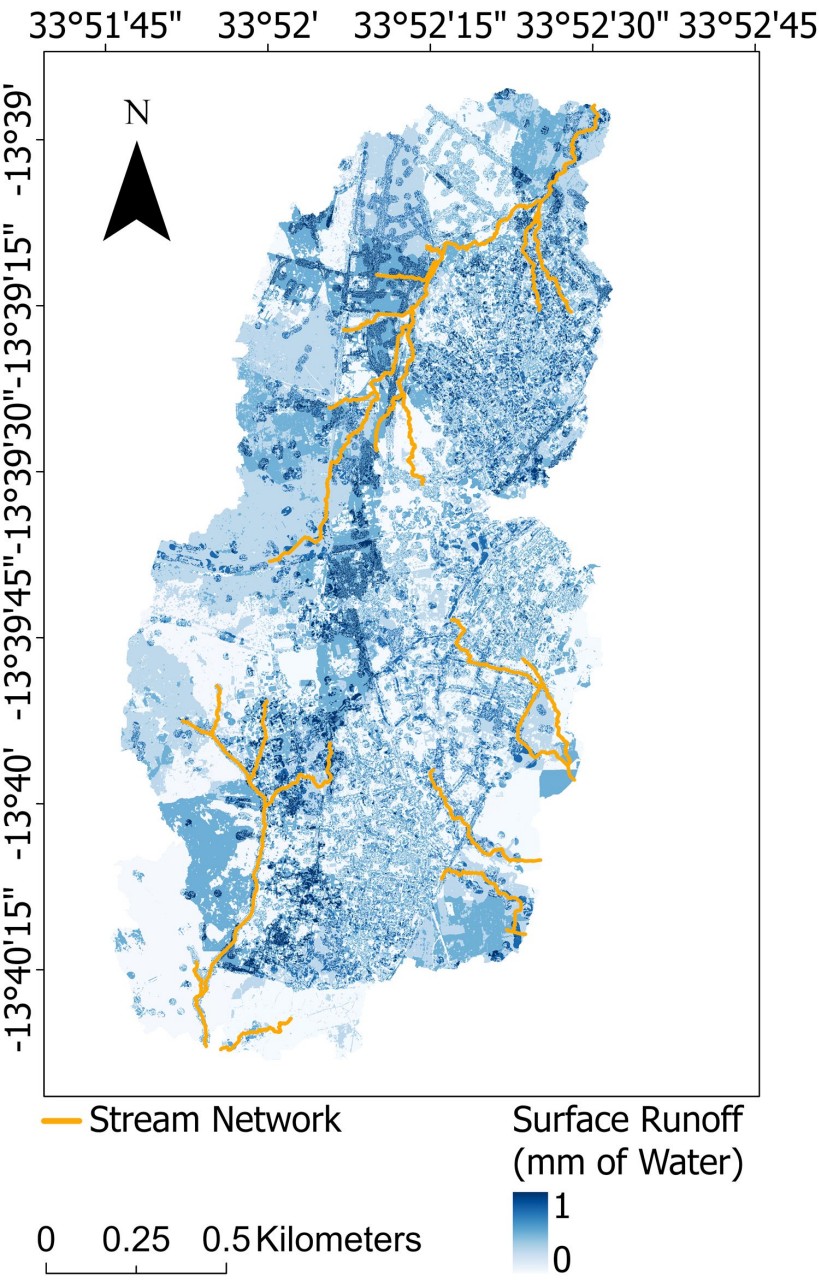

**Figure 17.** Comprehensive flood model of Dzaleka Refugee Camp, created by combining the normalized 10% exceedance probability of surface runoff and WCEP.

### 5.4. Relating Roof Type to Flood Risk

The comprehensive flood analysis is much more useful for urban planning and future building, but it is not incredibly helpful to the 41,000 refugees that already have homes. Drone imagery can provide some information about the conditions of a building based on the roof type. There are two types of roofs used at Dzaleka refugee camp: tin and thatched. Figure 18 provides an example of each of these.

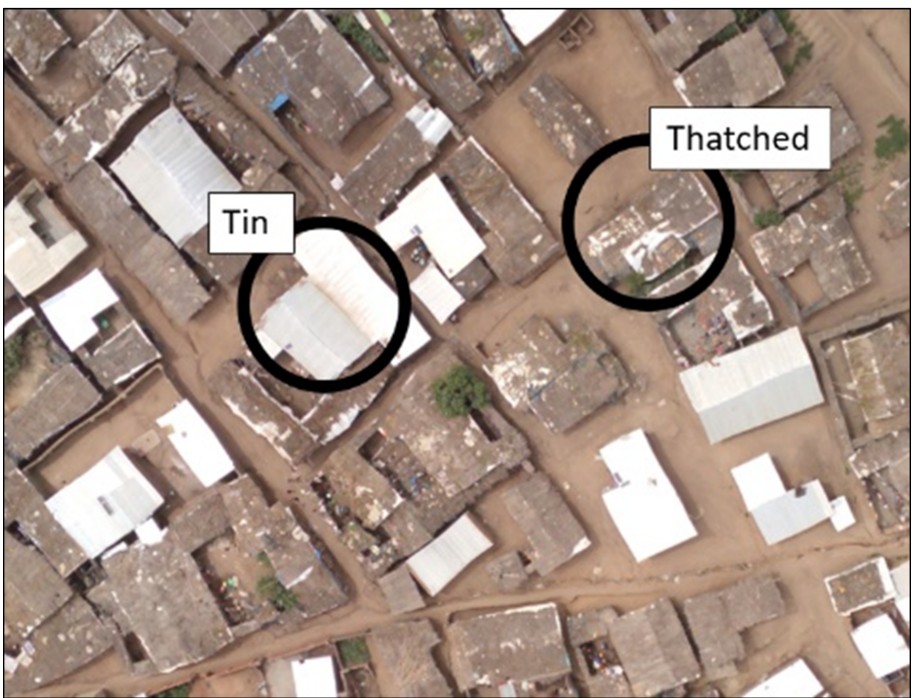

**Figure 18.** Examples of the two types of roofing at Dzaleka Refugee Camp: tin and thatched.

Minimal ground-truthing was completed at Dzaleka Refugee Camp to compare the risk of building collapse to building roof type. The risk of building collapse was given one of three qualitative ratings: little to no risk of failure, risk of failure, and large risk of failure. Although these are qualitative ratings, all buildings were evaluated by one person so the comparison between buildings should be consistent. Table 2 shares the findings from visually analyzing 15 buildings. More ground-truthing needs to be completed, but from Table 2 it is clear that thatched roofs have a higher risk of failure than tin roofs.

**Table 2.** Ground-truthing of 15 buildings, comparing roof type to risk of building collapse.

| Risk of Collapse | Thatched Roof | Tin Roof |
|---|---|---|
| Large Risk of Failure | 6 | 0 |
| Risk of Failure | 2 | 1 |
| Little to No Risk of Failure | 2 | 4 |

Using a model trained with Picterra, buildings were found and classified as either tin or thatched in the drone imagery (Figure 19a). Tin roofs (blue) were then given a weight of 0.2 while thatched roofs (red) were given a weight of 1. This weight was then multiplied by the comprehensive flood model (Figure 17) to find which buildings were at the highest risk. These weights were selected because of the initial ground-truthing results. As mentioned earlier, more ground-truthing still needs to be completed. Figure 19b maps the risk of each building collapsing from 0 to 1, with 1 being the most likely to collapse. The smaller study area explored in Section 5.3 is shown in Figure 20, where Figure 20a classifies the building type and Figure 20b maps the risk of building collapse.

Building risk was split into four categories: very low, low, high, and very high. Table 3 provides corresponding risk values and the percent of building area covered by each risk category. The approximate population living in a structure of each category was also calculated and shown in Table 3.

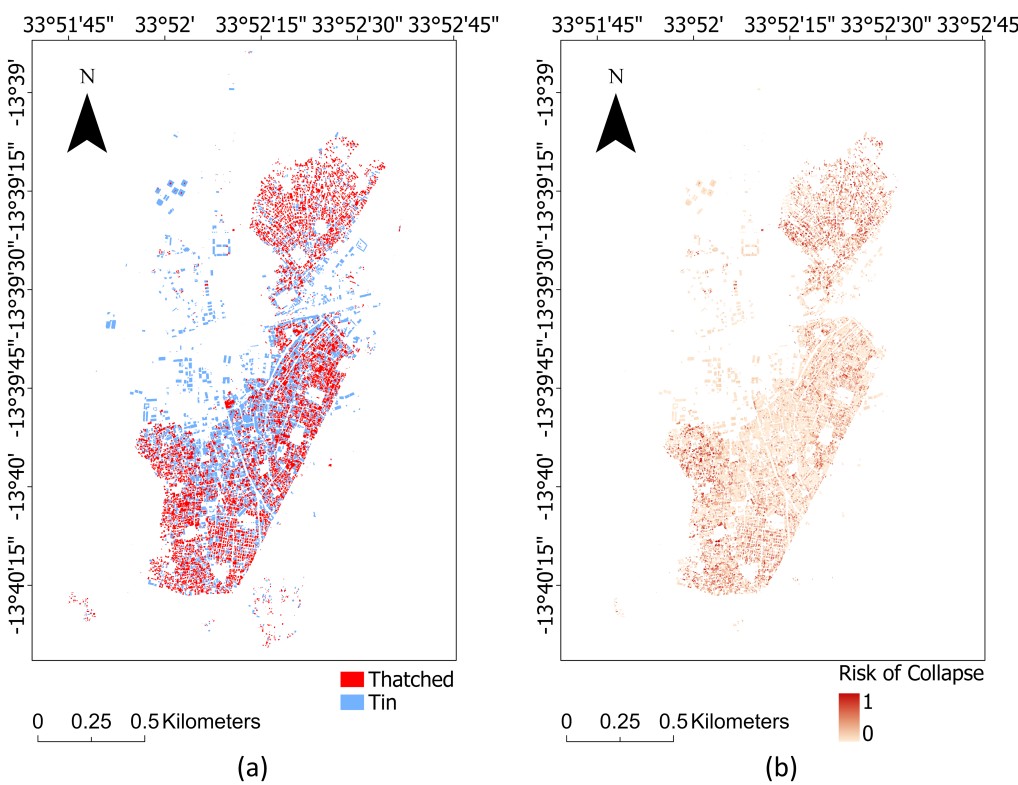

**Figure 19.** (**a**) Picterra found buildings classified as thatched (red) or tin (blue); (**b**) Building risk map normalized from 0 (white) to 1 (red), where 1 are locations of highest risk.

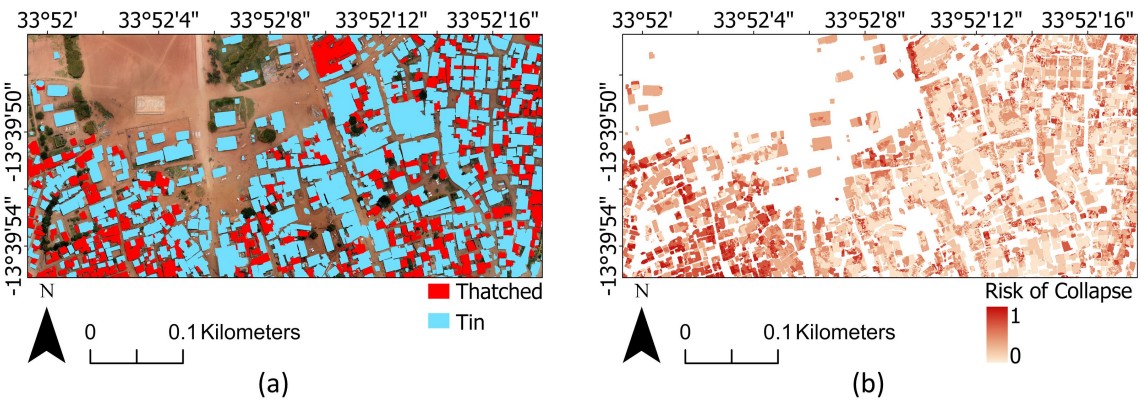

**Figure 20.** A small study area of Dzaleka Refugee Camp through building analysis processing: (**a**) buildings classified as thatched or tin; (**b**) building risk map created by combining building classifications (Figure 20a) and the comprehensive flood model (Figure 16f).

**Table 3.** Percentage of area covered for each risk category, along with the approximate population living there.

| Risk | Building Risk Value | % of Area | Population |
|---|---|---|---|
| Very Low | 0–0.25 | 70.19 | 28777 |
| Low | 0.25–0.5 | 13.66 | 5602 |
| High | 0.5–0.75 | 9.45 | 3874 |
| Very High | 0.75–1 | 6.70 | 2747 |

Even though more ground-truthing and validation needs to be completed for the study done at Dzaleka Refugee Camp, the results have the potential to support data-scarce

communities with a method for flood modeling and building risk (in relation to flood modeling) analysis. Using drone imagery to create a comprehensive flood model and then combining the comprehensive flood model with building type creates modeling potential for small areas that do not have any data.

## 6. Discussion and Conclusions

In this work, we present a concept of simplified data collection, 3D cadastre modeling, and natural disaster risk assessment through the use of low-cost drones and adapted open-source software. The complete solution architecture includes data collection with low-cost drones, 3D cadastre modeling with the drone data, and natural disaster risk quantification for buildings. To collect high-resolution aerial imagery, a small, low-cost, and autonomously flown drone was developed. The collected aerial imagery can be used to produce 3D cadastre models in order to contextualize and quantify potential natural disaster risk to infrastructure and property. The aerial imagery is first processed to produce georeferenced 3D reconstructions, digital elevation models, and orthomosaics. These can be combined with cadastre data in order to produce 3D cadastre models. While many studies have used 3D visualization and drone imagery with cadastre modeling and mapping [46,48,53,54], our work additionally processes the drone imagery to help determine risk data so that this can be contextualized with the 3D cadastre. In this work, the drone imagery was used as an input for flood modeling in order to produce risk data for a refugee camp. While our risk assessment work has focused on flood modeling, the collected data can be used to assess and contextualize many other potential natural disasters as well. Future work will need to be done to overlay the risk data on the 3D cadastre models. The combined risk and 3D cadastre data can help pinpoint exactly how specific buildings may be impacted. With flooding, for example, the 3D visualization would enable possible flood depth to be understood relative to the height of the buildings.

Using a drone, high-resolution aerial imagery was captured of the Dzaleka Refugee camp in Malawi. This aerial imagery was used to produce a digital terrain model and orthomosaic of the refugee camp. By classifying the land types in the orthomosaic, a land use map was created in order to be used as an input to the SWAT model. Despite these high-resolution data, the SWAT model was limited by the much lower resolution of the weather and soil data for the area. As a result, we combined the SWAT model with geomorphological information to create a comprehensive flood model. Lioi et al. [55] and Lastra et al. [56] both found that geomorphological information can enhance a flood model, especially in data-scarce areas. A machine learning program, Picterra, was then used to find erosion patterns in the drone imagery. The detected erosion patterns were compared to the flow direction of the DTM to determine which erosion patterns were WCEPs. A more comprehensive flood model was produced by combining the WCEP map and the 10% exceedance probability for surface runoff. The comprehensive flood model enables the inclusion of ground-truthed data using found WCEPs. Although this method still needs more verification, the resulting comprehensive flood model is very useful for future urban planning and has the potential to be applied to other data-scarce communities. Additionally, a flood damage metric can be scaled by property value estimates to produce individual and community property risk assessments.

In order to assess the condition of the buildings which are already in the refugee camp, the aerial imagery helped to examine the types of roofs on these buildings. The drone imagery was used to classify building types based on roof condition, which can help determine building integrity. Minimal ground-truthing was completed at the refugee camp to compare the risk of building collapse to the roof type, although more will still need to be completed. Using a machine learning model in Picterra, the buildings were found and classified based on their roof types, tin or thatched, in the drone imagery. Each roof type was given a weight and this was combined with the comprehensive flood model to determine the buildings at the highest risk. This produced building risk data which can be viewed and analyzed at both the community level and for individual properties.

This building risk data could then be combined with the 3D cadastre models in order to provide context to this data and present it in a format that is easy to view and understand for specific properties in a community. Additional future work will need to be completed to incorporate more ground-truth data and further validate these results, but the risk data are already extremely useful. This combination of the comprehensive flood model and building type analysis using drone imagery produces great risk modeling potential, particularly for data-scarce communities.

**Author Contributions:** Conceptualization, K.K. and J.W.; methodology, K.K.; software, D.W. and B.F.; validation, D.W. and B.F.; formal analysis, B.F.; investigation, D.W. and B.F.; resources, K.K., V.S., and J.W.; data curation, D.W. and B.F.; writing—original draft, D.W. and B.F.; writing—review and editing, K.K., V.S., and J.W.; visualization, D.W. and B.F.; supervision, K.K.; project administration, K.K.; funding acquisition, K.K. and J.W. All authors have read and agreed to the published version of the manuscript.

**Funding:** This research was funded through a grant from the National Science Foundation, Industry-University Research Cooperative Research Center, Center for Unmanned Aircraft System.

**Institutional Review Board Statement:** Not applicable.

**Informed Consent Statement:** Not applicable.

**Data Availability Statement:** The data presented in this study are available on request from the corresponding author.

**Acknowledgments:** We would like to thank the World Bank for their support. We would also like to thank UNICEF for allowing us to use their Parrot Disco and Parrot Sequoia to complete the flights at Dzaleka, Dzaleka Refugee Camp for allowing us to fly at the camp, and the African Drone and Data Academy Level 2 Cohort 1 Team for completing the flights.

**Conflicts of Interest:** The authors declare no conflict of interest.

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
