# Peer review of "Drone-Based Community Assessment, Planning, and Disaster Risk Management for Sustainable Development"

_remotesensing, doi:10.3390/rs13091739_

Round 1

Reviewer 1 Report

The paper deals with the relevant topics of community assessment, planning, and effective disaster risk management for sustainable development. Methodologically, the study is focused on a concept of a simplified data collection, and it includes data acquired through airborne laser-scanning (incl. low-budget drone data) and the modelling of 3D cadastral data.

This manuscript appears in a mature state. There are three points which I would like to mention. The author team should reflect and consider them in a revised version of the manuscript:

  • While the paper is well introduced, structured, written and methodologically sound, a severe problem occurs in the final chapter (“6. Discussion and Conclusions”): This rather short chapter in which the main findings should be present does not include any references to related studies (from the state-of-the-art literature). So, the readers do not get a compact presentation of the new contribution within this methodologically oriented study. Please re-write the end of your paper and highlight the key findings!

  • Some of the figures are not in a good quality for a final publication. They are acceptable for a review version but not for a published one. For example. Figure 18 suggests point clusters in violet. The legend (for polygons, not points!) can hardly be read. And the spatial excerpt in the base map seems to be a different one than in figure 19, 22, and 23. Could you change the scale to a larger scale which would make your maps more readable. Is it possible to avoid some of them, as your manuscript contains of 24 (!) figures. Please go through your figures, reduce the number and make maps more readable by changing the scale and font size in the legend.

  • In your methodological approach, the 3D aspect becomes a bit too short. The relevance of modern approaches to 3D cadastral modeling is not clarified. There are different approaches which are currently discussed in 3D visualization to enrich the possibilities of 3D (cadastral) mapping and modelling. For example, this refers to the flexibility of using UAVs for acquiring information on buildings, such as (historically relevant) landmarks which could be transferred to official public and governmentally maintained geospatial databases. There is also an ongoing debate on using 3D cadastral data to present this data in immersive virtual environments. The usage of 3D cadastral data in BIM applications is also under study and relevant for a future perspective. I would suggest pointing to these developments in your introductory sections to cover a bit more of the ongoing discussion and research directions in (cadastral) 3D modelling. There are some references I could recommend in these contexts:

https://doi.org/10.1007/s42489-020-00061-0

https://doi.org/10.1007/s42489-020-00069-6

https://doi.org/10.3390/ijgi8080329

Reviewer 2 Report

The study aims to present the research on “Drone-based Community Assessment, Planning, and Disaster Risk Management for Sustainable Development.” The manuscript is presented clearly and nicely. The Paper is interesting. However, there are some essential suggestions to improve the paper before acceptance. Thus, I would like to suggest major revisions. 

  1. Line 18- 35 citations are missing
  2. The objectives of the paper are missing in the introduction.
  3. Better to add the date of Figure 1
  4. The scale of Figure 20 is difficult to read
  5. You have many results, but the discussion is missing. Better to the strong discussion section and it will help maintain quality of the paper
  6. How this method can be used for similar study areas. Explain
  7. Some additional reading related to past studies is needed.

Reviewer 3 Report

The SWAT model should be calibrated, with missing data it is difficult to assess the reliability of the results. Secondly, the authors write about neural networks, while there is no information about the method of learning, testing, validation, about their structure. There is no data on how the data were selected for learning, testing, whether there was a validation? Figures 24, 22, 20 are not readable. Authors should organize their work and how they describe next threads not describe these results very randomly. 

Round 2

Reviewer 1 Report

The authors provided a new manuscript version in which all points of the first review round are addressed. The response letter is unusually short and does not include deeper thoughts on the changes made. However, the manuscript itself is in a mature version, and I would like to recommend this version for publication in this journal.

Author Response

Thank you very much for your review and feedback for our paper. Minor edits have been made after this round of reviews. We have made some changes regarding the English language and grammar throughout. Although we unfortunately do not have a change tracked version. The largest change was the addition of a sub-section (2.3. Machine Learning with Drone Imagery) to include background on the concept of CNNs and how we used Picterra. This was to address the comments from another reviewer.

Reviewer 2 Report

The paper is improved. But very difficult to capture changes without track changes version. In addition, I suggest copying relevant changes in the review response. 

Reviewer 3 Report

Dear Authors and Reviewers,
As, the authors pointed out that the methodology for data selection for learning, testing and validation is not provided. It seems to me that it is dangerous to test unverified tools by other authors. it seems to me that the manuscript should pay attention to technical details related to the issue of learning, testing and validation, because we should look for and use tools that are safe to use. If there are no details about the model (machine learning training with Picterra ) how can machine learning specialists use the results in the paper? For such people, knowing the structure of the model is crucial. 
The paper is obviously very interesting and the approach presented. An intermediate reader, not an expert in mathematical modeling may not be knowledgeable about the concept of CNNs, and I would ask for a brief concise introduction to the method.
